# Physical Inactivity and Sedentarism during and after Admission with Community-Acquired Pneumonia and the Risk of Readmission and Mortality: A Prospective Cohort Study

**DOI:** 10.3390/jcm11195923

**Published:** 2022-10-07

**Authors:** Camilla Koch Ryrsø, Arnold Matovu Dungu, Maria Hein Hegelund, Daniel Faurholt-Jepsen, Bente Klarlund Pedersen, Christian Ritz, Birgitte Lindegaard, Rikke Krogh-Madsen

**Affiliations:** 1Department of Pulmonary and Infectious Diseases, Copenhagen University Hospital—North Zealand, 3400 Hillerød, Denmark; 2Centre for Physical Activity Research, Copenhagen University Hospital—Rigshospitalet, 2100 Copenhagen, Denmark; 3Department of Infectious Diseases, Copenhagen University Hospital, Rigshospitalet, 2100 Copenhagen, Denmark; 4National Institute of Public Health, University of Southern Denmark, 1455 Copenhagen, Denmark; 5Department of Clinical Medicine, University of Copenhagen, 2200 Copenhagen, Denmark; 6Department of Infectious Diseases, Copenhagen University Hospital, Hvidovre, 2650 Copenhagen, Denmark

**Keywords:** community-acquired pneumonia, hospital admission, length of stay, mortality, readmission, physical activity

## Abstract

Background: Bed rest with limited physical activity is common during admission. The aim was to determine the association between daily step count and physical activity levels during and after admission with community-acquired pneumonia (CAP) and the risk of readmission and mortality. Methods: A prospective cohort study of 166 patients admitted with CAP. Step count and physical activity were assessed with accelerometers during and after admission and were categorised as sedentary, light, or moderate-vigorous physical activity. Linear regression was used to assess the association between step count and length of stay. Logistic regression was used to assess the association between step count, physical activity level, and risk of readmission and mortality. Results: Patients admitted with CAP were sedentary, light physically active, and moderate-to-vigorous physically active 96.4%, 2.6%, and 0.9% of their time, respectively, with 1356 steps/d. For every 500-step increase in daily step count on day 1, the length of stay was reduced by 6.6%. For every 500-step increase in daily step count during admission, in-hospital and 30-day mortality was reduced. Increased light and moderate-to-vigorous physical activity during admission were associated with reduced risk of in-hospital and 30-day mortality. After discharge, patients increased their daily step count to 2654 steps/d and spent more time performing light and moderate-to-vigorous physical activity. For every 500-step increase in daily step count after discharge, the risk of readmission was reduced. Higher moderate-to-vigorous physical activity after discharge was associated with a reduced risk of readmission. Conclusions: Increased physical activity during admission was associated with a reduced length of stay and risk of mortality, whereas increased physical activity after discharge was associated with a reduced risk of readmission in patients with CAP. Interventions focusing on increasing physical activity levels should be prioritised to improve the prognosis of patients admitted with CAP.

## 1. Introduction

Community-acquired pneumonia (CAP) remains a leading cause of hospital admission, with one in five patients being readmitted within 30 days after discharge [1,2]. Bed rest with limited physical activity is common during admission. In patients with CAP, both external (e.g., oxygen therapy, intravenous antibiotic treatment) and internal (e.g., fatigue, hypoxemia) factors limit physical activity during admission [3]. As a result, patients admitted with CAP spend over 90% of their in-hospital time being physically inactive, with 900–1300 steps/d during admission [4,5]. In comparison, the average number of daily steps is approximately 6500 steps/d for healthy individuals [6]. Physical inactivity is not only a concern among patients with low functional status; despite the ability to walk independently at admission, only 28% of medical patients admitted with respiratory, gastrointestinal, or renal disease walked during admission [7]. In patients with CAP, a lower daily step count during admission has been associated with a prolonged length of stay [4]. In addition, physical inactivity during admission has been associated with readmission and mortality in older medical patients admitted with respiratory, cardiovascular, or gastrointestinal diseases [8,9]. Hospital-associated deconditioning with loss of muscle mass and strength is a serious concern, as up to 40% of older patients lose the ability to perform activities of daily living after discharge [10,11,12]. For 40% of these patients, these newly acquired disabilities will never recover [12]. However, physical activity is a central and potentially modifiable factor to prevent hospital-associated functional decline and adverse outcomes in older medical patients [10,11]. In patients with CAP, an exercise intervention initiated during admission can, to some extent, counteract the negative consequences of physical inactivity on muscle strength and the ability to perform daily activities [13]. To our knowledge, no previous study has investigated the association between 24-h physical activity levels during and after admission on the prognosis in patients admitted with CAP.

We hypothesised that physical inactivity and sedentarism during admission and immediately after discharge were associated with an increased risk of severe outcomes in patients admitted with CAP. We aimed to determine 24-h physical activity levels and the daily step count during admission and immediately after discharge in patients with CAP and the association with prognosis, such as length of stay, 30-day readmission, and in-hospital and 30-day mortality.

## 2. Methods

### 2.1. Study Design, Settings, and Study Population

This study is part of the Surviving Pneumonia Cohort Study, a prospective cohort study including patients admitted with CAP at the Copenhagen University Hospital—North Zealand, Denmark. Inclusion criteria were age ≥18 years and suspected CAP defined as a new pulmonary infiltrate on chest X-ray or computed tomography scan and minimum 1 symptom consistent with CAP, e.g., fever (≥38.0 °C), hypothermia (<35.0 °C), cough, sputum production, pleuritic chest pain, dyspnea, or focal chest signs on auscultation. Exclusion criteria for the present study were no attachment of accelerometer at admission, paralysis of legs (non-ambulant), expected length of stay ≤48 h, or less than one day with ≤20 h physical activity data during admission. Patients were enrolled within the first 24 h of admission. Patients in the present study were included between January 2019 and April 2022.

### 2.2. Study Variables

Data were collected prospectively with standardised forms and were entered into a REDCap database. Information about demography, prior medical history, comorbidities, and clinical outcomes were collected during an interview at the study enrolment and from medical records. The CURB-65 score [14] was used to risk-stratify patients and was classified as mild (score 0–1), moderate (score 2), or severe (score 3–5) CAP. The combined burden of comorbidities was assessed by the Charlson Comorbidity Index [15] and categorised as 0, 1, or ≥2 comorbidities. Data upon admission to the intensive care unit (ICU), length of stay, readmission, and mortality were collected from medical records up to 30 days after discharge.

The self-reported physical activity level prior to admission was assessed with the short form of the international physical activity questionnaire (IPAQ) [16]. Patients were categorised into 3 levels of physical activity (low <600 metabolic equivalents of task (MET)-min/week, moderate ≥600 MET-min/week, or high ≥3000 MET-min/week) [17].

Physical activity levels were objectively assessed using the Axivity AX3 accelerometer. The accelerometers were initialised to measure at 100 Hz with ±8 g bandwidth using the Open Movement software (OmGui, version 1.0.0.43, Newcastle University, Newcastle upon Tyne, UK). The accelerometer was attached directly to the skin on the medial front of the right thigh, midway between the hip and knee joints, with its positive *x*-axis pointing inferiorly and its negative *z*-axis pointing anteriorly [18]. A 50 × 100 mm section of Mefix tape (Mölnlycke Health Care, Göteborg, Sweden) with a stripe of double-sided adhesive tape was placed on top of the clean, dry skin. The accelerometer was placed on the double-sided tape and secured to the site with a 110 × 140 mm piece of transparent film (Leukomed T, BNS medical GmbH, Hamburg, Germany). Patients were instructed to wear the accelerometer for up to 7 consecutive 24-h periods during admission or until discharge (if discharged before day 8) and 7 consecutive 24-h periods after discharge. Data were collected between 05:00 AM on day 1 and 05:00 AM on day 8.

Data from the accelerometers were downloaded in the original cwa file format using the OmGui software and converted to a binary gt3x compatible file format using a custom-made add-on to OmGui to assess intensity estimates using ActiLife (version 6.13.4, ActiGraph, Pensacola, FL, USA). The accelerometer wear time was determined manually using OmGui based on raw accelerometry. Data were extracted in 1 s epochs. Non-wear time was defined as ≥180 consecutive min of zero counts/min, allowing for up to 2 min of non-zero counts if the interruption was preceded or followed by ≥30 min of zero counts/min. A valid 24-h measurement was defined as at least 20 h of wear time. Patients were included if they had at least one day with ≥20 h out of 24 h of wear time [19]. Daily step counts were calculated from the step detection algorithm in ActiLife using the recordings of raw accelerations from the 3 axes [20]. Sedentary time was defined as time spent at ≤100 counts/min [21], light physical activity as 100–1951 counts/min, and moderate-to-vigorous physical activity as ≥1952 counts/min [22].

### 2.3. Outcomes

The primary outcome was a 30-day readmission, with secondary outcomes being length of stay and in-hospital and 30-day mortality. The days from discharge to readmission and cause of readmission were registered. Patients with multiple readmissions were registered with reference to the first readmission. All variables were collected from medical records. Age, sex, CURB-65, ICU admission, mechanical ventilation, non-invasive ventilation, and high flow therapy were viewed as potential confounders and were adjusted for. No stepwise selection process was applied. 

### 2.4. Statistical Analysis

Data were described as counts (%) for categorical variables and either means (standard deviation (SD)) or medians (interquartile range (IQR)) for continuous variables as appropriate. Binary logistic regression analyses were used to assess the association between daily step count and physical activity levels (time spent in sedentary behaviour, light physical activity, and moderate-to-vigorous physical activity) and risk of 30-day readmission and in-hospital and 30-day mortality, respectively. Both unadjusted, univariate models and adjusted, multivariate models were fitted. To adjust for confounding, analyses included age, sex, CURB-65, ICU admission, mechanical ventilation, non-invasive ventilation, and high flow therapy. Linear regression was used to determine the association between step count on day 1 with the length of stay. Multivariate models included age, sex, and CURB-65. Model assumptions, including normality, were assessed using residual and quantile-quantile plots. Due to the skewed distribution of length of stay, the variable was log-transformed, and the regression coefficients were back-transformed to provide ratios. Wilcoxon signed-rank tests were used to detect differences in daily step counts and physical activity levels from admission to after discharge. Chi-squared tests were used to explore differences in physical activity levels prior to admission between patients included before compared to during or after the COVID-19 lockdown. All *p*-values were two-sided, and significance levels were *p* < 0.05. Data were analysed using IBM SPSS Statistics version 25.

### 2.5. Research Ethics

Patients provided informed consent before enrolment. The study was approved by the Scientific Ethics Committee at the Capital Region of Denmark (H-18024256), registered on ClinicalTrials.gov (NCT03795662), and conducted according to the Declaration of Helsinki [23]. This reporting of the study followed the Strengthening the Reporting of Observational Studies in Epidemiology statement [24].

## 3. Results

Initially, 189 patients were included in the study; however, 23 patients were excluded from the analysis due to missing physical activity data from admission, leaving 166 patients with physical activity data during admission. After discharge, 89 patients were excluded from the analysis due to missing physical activity data, leaving 77 patients with physical activity data after discharge to be included in the analysis (Figure 1).

### 3.1. Demography, Comorbidities, and Clinical Parameters

Patient characteristics are summarised in Table 1. The median age of the study population was 75 years, with 71.1% of the patients being 65 years of age or older. One hundred patients (60.2%) had ≥2 comorbidities, and 78 (50.3%) had mild CAP. Based on IPAQ, 80.1% of the patients had a low physical activity level prior to admission, while 13.2 and 6.6% had a moderate or high physical activity level (Table 1). There was no difference in physical activity levels prior to admission between patients included before the COVID-19 lockdown (January 2019–February 2020) and patients included during or after the COVID-19 lockdown (March 2020–April 2022, Appendix A).

During admission, 156 patients (94.0%) received intravenous antibiotic treatment, and 117 (70.5%) were treated with supplementary oxygen. During admission, five patients (3.0%) were admitted to the ICU, and 13 (7.8%) died. Sixteen patients (9.6%) died within 30 days after discharge. The median length of stay was 7 days. Thirty-seven patients (24.2%) were readmitted within 30 days after discharge, with pulmonary conditions (e.g., CAP, acute exacerbation of chronic obstructive pulmonary disease (COPD)) being the most common causes of readmission (51.4%, Appendix A).

### 3.2. Physical Activity and Sedentary Behaviour during Admission and after Discharge

The mean wear time of the accelerometers was 4.0 ± 2.2 days during admission (45.4 ± 18.5% of admission time) and 6.3 ± 1.6 days after discharge (90.0 ± 22.4% of follow-up time). During admission, patients spent 96.4% of their time in sedentary behaviour, 2.6% in light physical activity, and 0.9% in moderate-to-vigorous physical activity. The median daily step count during admission was 1356 steps/d. After discharge, patients increased their daily step count to 2654 steps/d and spent more time in light and moderate-to-vigorous physical activity (4.2% and 1.8%, Table 2).

### 3.3. Association between Physical Activity and Sedentary Behaviour during Admission and Prognosis

In a multivariable analysis adjusted for age, sex, and CURB-65, length of stay was reduced by 6.6% (95% CI 2.0–10.9%; *p* < 0.01) for every 500-step increase in daily step count on day 1. There was no association between daily step count or time spent in sedentary behaviour, light physical activity, and moderate-to-vigorous physical activity during admission and risk of 30-day readmission (Figure 2).

Risk of in-hospital and 30-day mortality was reduced between 44–48% for every additional 500-step increase in daily step count during admission (Figure 2). Further adjustment with ICU admission, mechanical ventilation, non-invasive ventilation, and high flow therapy did not change the estimates, but models were no longer significant. Risk of in-hospital and 30-day mortality increased between 55–63% for every 1-percentage point increase in sedentary behaviour during admission, corresponding to a 14-min increase in sedentary behaviour (Figure 2). Further adjustment with ICU admission, mechanical ventilation, non-invasive ventilation, and high flow therapy did not change the estimates, but models were no longer significant.

Risk of in-hospital and 30-day mortality decreased between 47–51% for every 1-percentage point increase in light physical activity during admission (Figure 2). Further adjustment with ICU admission, mechanical ventilation, non-invasive ventilation, and high flow therapy did not change the estimates, but models were no longer significant.

Risk of in-hospital and 30-day mortality decreased between 73–79% for every 1-percentage point increase in moderate-to-vigorous physical activity during admission (Figure 2). Further adjustment with ICU admission, mechanical ventilation, non-invasive ventilation, and high flow therapy did not change the estimates, but models were no longer significant.

### 3.4. Association between Physical Activity and Sedentary Behaviour after Discharge and Prognosis

Risk of 30-day readmission was reduced between 21–24% for every additional 500-step increase in daily step count after discharge (Figure 3).

Risk of 30-day readmission increased by 24–29% for every 1-percentage point increase in sedentary behaviour after discharge (Figure 3). There was no association between time spent in light physical activity and risk of 30-day readmission.

Risk of 30-day readmission decreased by 54–63% for every 1-percentage point increase in moderate-to-vigorous physical activity after discharge (Figure 3).

There was no association between daily step count or time spent in sedentary behaviour, light physical activity, and moderate-to-vigorous physical activity after discharge and risk of 30-day mortality (Figure 3).

## 4. Discussion

We examined the impact of physical activity levels during admission and immediately after discharge on the prognosis in patients with CAP. Overall, patients admitted with CAP spend most of their time in sedentary behaviour with a low daily step count. After discharge, patients increase their daily step count and engage in more light and moderate-to-vigorous physical activity. First, our results showed that increased daily step count on the first day of admission is associated with a reduced length of stay. Second, increased physical activity and less time spent in sedentary behaviour during admission are associated with a reduced risk of in-hospital and 30-day mortality. Third, increased physical activity and less time spent in sedentary behaviour after discharge are associated with a reduced risk of 30-day readmission. Finally, our findings demonstrate the crucial association between physical activity during admission and immediately after discharge on prognosis in patients with CAP.

Bed rest with limited physical activity is common during admission. Patients admitted with CAP spent over 96% of their in-hospital time in sedentary behaviour, with only 1356 steps/d, which is similar to previous observations [4,5]. Further, we showed that, for every 500-step increase in daily step count on the first day of admission, the length of stay was reduced by 6.6%, corresponding to 0.5 days reduction. Our findings are consistent with a recent study by Rice and colleagues [4] showing an 11% reduced length of stay, corresponding to 0.4 days reduction, for every 500-step increase in daily step count during admission in patients with CAP.

An increased daily step count during admission was associated with a 44–51% reduced risk of in-hospital and 30-day mortality. In comparison, a previous study of older medical patients admitted with respiratory, cardiovascular, or gastrointestinal diseases showed a 2% reduced risk of 2-year mortality with an increased daily step count in the first 24 h of admission [9]. Similarly, a decline in daily step count from the first to the last 24 h of admission was associated with a 4-fold increased risk of 2-year mortality [9]. Further, we showed that an increase in light physical activity and a concurrent reduction in sedentarism during admission was associated with a 47–53% reduced risk of in-hospital and 30-day mortality. While physical activity is associated with an improved prognosis, an important issue to address is whether the low physical activity levels seen in patients admitted with CAP are simply a marker of disease severity, where severely ill patients are predominantly sedentary, and thus their poor disease outcomes should be attributed to their disease severity alone rather than a combination of disease severity and sedentarism. In a secondary analysis of the association between physical activity during admission and mortality, we included multiple variables (ICU admission, mechanical ventilation, non-invasive ventilation, and high flow therapy) to control for disease severity. After these adjustments, the estimates for the association between physical activity during admission and risk of in-hospital and 30-day mortality remained the same, but models were no longer significant.

We did not find any association between daily step count during admission and risk of 30-day readmission, which is in line with earlier observations from patients with CAP [4]. However, a 10% reduced risk of 30-day readmission has previously been reported in older medical patients admitted with respiratory, cardiovascular, or gastrointestinal diseases with an increased daily step count during admission [8].

Similar to previous studies measuring physical activity levels in patients admitted with CAP, we included patients aged ≥18 years [4,5]. Though the risk of CAP-requiring admissions increases with age, younger patients (≤64 years) admitted with CAP are often characterised with multiple comorbidities, which increases the risk of disease severity leading to prolonged length of stay and bed rest [25]. Even though earlier bed rest studies have shown that older healthy individuals are more susceptible to loss of muscle than younger during short-term bed rest [26], a substantial loss of muscle mass and strength still occurs in younger individuals [27,28].

Physical inactivity is not only a concern during admission but proceeds long after discharge. In the week following discharge, the daily step count significantly increased to 2654 steps/d, which is similar to previous observations from patients with CAP [5]. Further, we showed that increased daily step count and moderate-to-vigorous physical activity after discharge were associated with a 21–63% reduced risk of 30-day readmission in patients with CAP. Similarly, in a study of older medical patients admitted with respiratory, cardiovascular, or gastrointestinal diseases, increased daily step count the week following discharge was associated with a 15% reduced risk of 30-day readmission [29].

We did not find any association between the physical activity level after discharge and risk of 30-day mortality, as only 2 out of 77 patients died during the 30-day follow-up period from discharge. However, we still believe it is important to report odds ratios and confidence intervals to provide a quantification of the magnitude of the association. Results from our study could indicate that physical activity during admission is more related to survival, whereas physical activity after discharge is associated with recovery after discharge and whether readmissions can be prevented. Although physical inactivity often reflects disease severity, encouraging physical activity in all disease stages could be a powerful strategy to reduce the risk of readmission.

After discharge, the patients spent less time in sedentary behaviour than during admission, which is in line with observations from previous studies of patients with CAP [5,30]. However, in contrast to a study by Clausen and colleagues [5] of patients admitted with CAP, patients in our study spent considerably more time in sedentary behaviour after discharge (20.2 h/d vs 22.5 h/d). Differences in inclusion criteria between the studies could explain the different findings, as Clausen and colleagues [5] excluded patients who were admitted to the ICU within 24 h of admission. In contrast, we included patients despite ICU admission. Indeed, patients admitted to the ICU might display more sedentarism and delayed increase in physical activity level after discharge as a consequence of critical illness [31]. Our results could suggest that promoting physical activity through intervention may be required to increase physical activity levels during admission and after discharge. However, randomised controlled trials comparing the effect of standard of care combined with exercise training to standard of care alone are needed to explore whether an increased physical activity level during admission and after discharge improves the prognosis among patients with CAP.

Our findings emphasise the powerful association between the prognosis and interruption of sedentarism with increased physical activity levels in patients with CAP both during admission and immediately after discharge. It is, however, not known if low physical activity is a marker or a cause of severe disease. Nonetheless, hospitals play a crucial role in providing opportunities for patients to be physically active despite disease stages and acute illness. The prescription of exercise as a standardised part of the treatment for CAP could motivate patients to be more physically active even after discharge from the hospital. Indeed, previous studies of patients with CAP have shown that early and progressive mobilisation initiated within the first 24 h of admission reduces the length of stay by 1.1 to 2.1 days without increasing the risk of adverse events [32,33]. However, no study has to date looked at the effects of exercise training initiated during admission on the prognosis in patients with CAP.

Nonetheless, exercise training initiated immediately after discharge in patients admitted with acute exacerbation of COPD has been shown to reduce the number of days in the hospital and the risk of readmission and mortality [34]. These previous findings suggest that increasing physical activity levels through exercise interventions could be a way to improve the prognosis of patients admitted with CAP. Further results from our study suggest that relatively small increases in physical activity levels, both during admission and immediately after discharge, are associated with significant improvements in the prognosis of patients with CAP.

### Strength and Limitations

The strength of this study is the large sample size, the broad inclusion criteria (i.e., including severely ill patients admitted to the ICU and patients who used walking aids), the use of validated accelerometers to assess physical activity both during admission and after discharge, and the large number of activity hours recorded (15,741 h during admission and 11,640 h after discharge). Further, we add to the current evidence by showing low physical activity levels in patients admitted with CAP. However, this is the first study showing the association between low physical activity levels and the increased risk of readmission and mortality in patients with CAP.

Our study had some limitations. First, there is a risk of selection and inclusion bias, as informed consent had to be obtained within the first 24 h of admission. Therefore, we cannot rule out that patients’ refusal to participate in our study could be associated with the severity of their disease. Second, we excluded patients with a length of stay ≤48 h and ≤20 h of valid physical activity measurement. This might have led to the exclusion of patients with higher physical activity levels and less disease severity. However, we measured patients as many days as logistically possible and only included patients in the analysis if they had at least 1 day with ≥20 h out of 24 h of wear time to ensure both day and night measurement. Third, we only had physical activity data after discharge from 77 out of the 153 patients who were discharged alive, as patients refused to wear the accelerometer after discharge. However, there were no differences in age, sex, Charlson comorbidity index, number of comorbidities, CURB-65, or physical activity levels during admission between patients who wore an accelerometer after discharge and those who refused to wear it. Therefore, we assume that the 77 patients who wore the accelerometer after discharge are representative of the whole study population.

## 5. Conclusions

The present study provides evidence that physical inactivity and sedentarism are correlated with prolonged length of stay and increased risk of readmission and mortality in patients admitted with CAP. However, randomised controlled trials are needed to establish whether exercise training initiated during admission improves prognosis among patients with CAP.

## Figures and Tables

**Figure 1 jcm-11-05923-f001:**
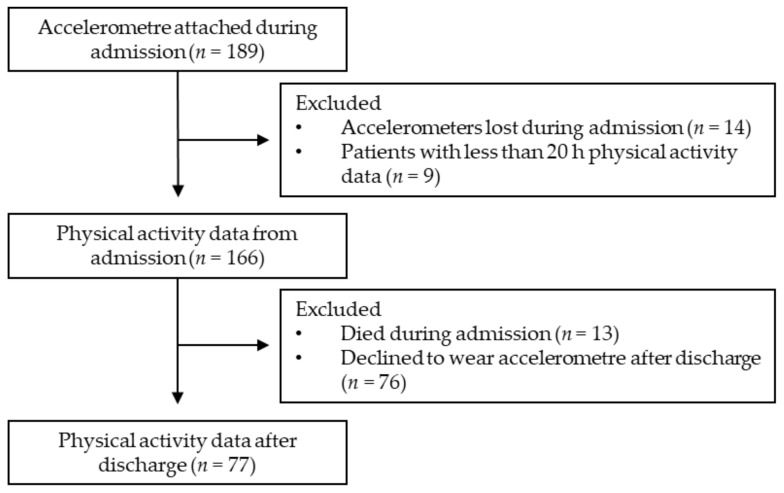
Flow chart of the study population.

**Figure 2 jcm-11-05923-f002:**
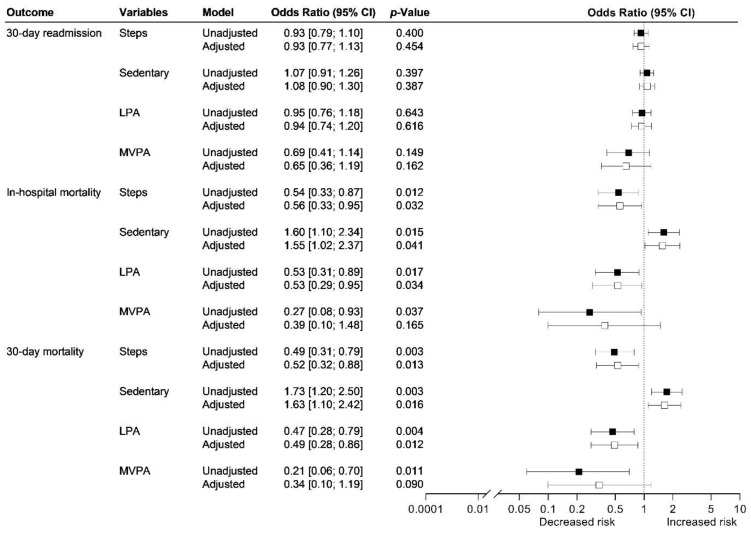
Association between physical activity and sedentary behaviour during admission and risk of readmission and mortality among 166 patients admitted with community-acquired pneumonia. The association between daily step count and physical activity level during admission and risk of 30-day readmission, in-hospital mortality, and 30-day mortality were analysed using unadjusted and adjusted logistic regression models (adjustments: age, sex, and CURB-65 (white)). Steps: Odds ratio per 500-step increase in daily step count during admission. The patients who died in the hospital are not included in the analyses for readmission. LPA: light physical activity, MVPA: moderate-to-vigorous physical activity.

**Figure 3 jcm-11-05923-f003:**
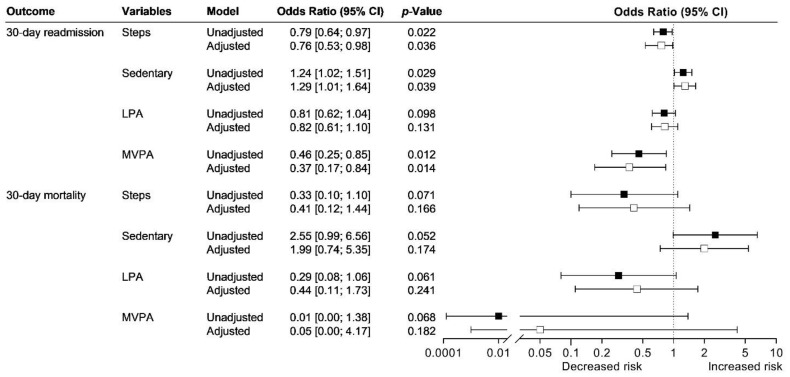
Association between physical activity and sedentary behaviour after discharge and risk of readmission and mortality among 77 patients admitted with community-acquired pneumonia. The association between daily step count and physical activity level after discharge and risk of 30-day readmission and mortality were analysed using unadjusted (black) and adjusted logistic regression models (adjustments: age, sex, and CURB-65 (white)). Steps: Odds ratio per 500-step increase in daily step count after discharge. LPA: light physical activity, MVPA: moderate-to-vigorous physical activity.

**Table 1 jcm-11-05923-t001:** Baseline characteristics of 166 patients admitted with community-acquired pneumonia.

	Study Population (*n* = 166)
Age, median (IQR), years	75 (63–81)
Sex, male, *n* (%)	90 (54.2)
Charlson comorbidity index, median (IQR)	5 (3–6)
Number of comorbidities, *n* (%)	
0	24 (14.5)
1	42 (25.3)
≥2	100 (60.2)
Chronic obstructive pulmonary disease	62 (37.3)
Other chronic respiratory diseases	37 (22.3)
Malignancy	33 (19.9)
Diabetes	31 (18.7)
Chronic heart failure	30 (18.1)
Other chronic heart diseases	90 (54.2)
Cerebrovascular disease	26 (15.7)
Chronic kidney disease	7 (4.2)
Chronic liver disease	4 (2.4)
CURB-65	
0–1, *n* (%)	78 (50.3)
2, *n* (%)	57 (36.8)
3–5, *n* (%)	20 (12.9)
Physical activity level prior to admission	
Low, *n* (%)	121 (80.1)
Moderate, *n* (%)	20 (13.2)
High, *n* (%)	10 (6.6)
Clinical outcome	
Intravenous antibiotic treatment, *n* (%)	156 (94.0)
Oxygen therapy, *n* (%)	117 (70.5)
Intensive care unit, *n* (%)	5 (3.0)
Length of stay, median (IQR), days	7.2 (5.2–12.4)
In-hospital mortality, *n* (%)	13 (7.8)
30 days mortality, *n* (%)	16 (9.6)
30 days readmission, *n* (%)	37 (24.2)
CURB-65: confusion, urea, respiratory rate, blood pressure, and age ≥65 years. Missing variables: physical activity level (*n* = 15, 9.0%), CURB-65 (*n* = 11, 6.6%).	

**Table 2 jcm-11-05923-t002:** Physical activity and sedentary behaviour during admission and after discharge among 77 patients admitted with community-acquired pneumonia.

	During Admission (*n* = 77)	After Discharge (*n* = 77)	*p*-Value
Time spent in sedentary behaviour, median (IQR), %	96.0 (94.8–96.9)	93.8 (91.8–95.7)	<0.001
Time spent in light physical activity, median (IQR), %	3.1 (2.2–3.9)	4.2 (2.9–6.1)	<0.001
Time spent in moderate-to-vigorous physical activity, median (IQR), %	1.0 (0.7–1.5)	1.8 (1.1–2.8)	<0.001
Steps, median (IQR), *n*/day	1588 (1130–2226)	2654 (1810–3847)	<0.001

Comparisons were made with Wilcoxon signed-rank tests.

## Data Availability

Datasets used for the current study are not publicly available. However, relevant pseudonymised data can be accessed upon a reasonable request to the corresponding author. The lead author affirms that the manuscript is an honest, accurate, and transparent account of the study being reported; that no important aspects of the study have been omitted; and that any discrepancies from the study as planned have been explained.

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
