# Peer review of "Physical Inactivity and Sedentarism during and after Admission with Community-Acquired Pneumonia and the Risk of Readmission and Mortality: A Prospective Cohort Study"

_jcm, 2022, doi:10.3390/jcm11195923_

Round 1

Reviewer 1 Report

Overall a relevant study, that shows an assocation between mobility and outcome in pneumonia.  - Needs major revision and clarity and more caution in the conclusions drawn from the data

Introduction – separate into paragraphs by concept

How were covariates chosen and what kind of logistic model was used binary, ordinal, and if binary was a stepwise selection applied. More info needed in methods section on outcomes.

High drop out rate because of missing step data,  and exclusion of < 48 hours could  favour exclusion of more mobile patients - needs to be mentioned later in the discussion

Sedentary behaviour predominant in group

Note labelling for figures 2 and 3 is confusing and it is difficult to explain why different numbers for the same variables are noted between figure one and two

Figure 2.

No impact on 30 day readmission  as adjusted rates cancel the effect because of the possible impact of other co morbidities. There is however a significant effect on in hospital and 30 mortality which is only attenuated by adjustment to comorbities

Figure 3.

After discharge there is an effect on re-admisson but not on 30 day mortality as these seems to be again to be attenuated by comorbidities

In the model the steps variable and the sedentary variable may be the ssame variable and seem to mirror each other on the opposite side of the axis,

Discussion must be qualified

Overall the discussion is confusing, where the main findings form the studies and interspersed with reference from the literature. For clarities’ sake, first summarize the positive findings and then progressively show which literature support or disputes such conclusions.

The paragraphs are too long and should be more concise explaning one point/finding at a time.

While increased exercise may result in a better prognosis, it is also possible that the patients who were less mobile were probably more sick from the pneumonia and the systemic effects of bacteraemia/ septicaemica made them more lethargic more lethargic and as a result more likely to have a worse prognosis. This must be clearly stated in the discussion

Over here I see a prospective observational study, while to prove such a hypothesis a randomized trial of standard therapy vs standard therapy plus exercise would be necessary, something which is difficult to organise in acutely sick patients.

                             In addition, we showed that the risk of in-hospital and 30-day mortality 241 was reduced with an increase in light physical activity and a concurrent reduction in sed- 242 entarism during admission in patients with CAP

The authors imply a cause and effect relationship , while this study shows an assocaiation between exercise and better outcome, something which can be explained that more mobile patients are simply less sick. The effect of adjustment on the data tends to favour more the second possibility rather then the first.

Our results suggest 267 that promoting physical activity through intervention may be required to increase physi- 268 cal activity levels during admission and after discharge to improve the prognosis among 269 patients with CAP.

While the word “suggest” and “may”  is appropriate it must be clear to the readers, that interventional studies are necessary to prove such a hypothesis.

Conclusion

These findings highlight the need to implement exercise 312 interventions to increase physical activity levels to test the impact on the prognosis of 313 patients admitted with CAP.

Disagree with this statement

I think what the authors mean is that  interventional studies are needed to support such a hypothesis.

Author Response

Overall a relevant study, that shows an assocation between mobility and outcome in pneumonia. 

- Needs major revision and clarity and more caution in the conclusions drawn from the data.

Introduction – separate into paragraphs by concept

  • A paragraph has been added to the introduction (see line 66).

How were covariates chosen and what kind of logistic model was used binary, ordinal, and if binary was a stepwise selection applied. More info needed in methods section on outcomes.

  • A new section, "Outcomes," has been added to the manuscript (line 124-131). We used binary logistic regression analysis. Information about the model has been added to the manuscript (line 135).

High drop out rate because of missing step data,  and exclusion of < 48 hours could  favour exclusion of more mobile patients - needs to be mentioned later in the discussion

  • We acknowledge that the exclusion criteria with a length of stay ≤48 hours might have led to the exclusion of patients with higher physical activity levels and less severity of CAP. However, due to practical reasons, it was not possible to include patients with a shorter length of stay as patients were included within the first 24 hours of their admission, and the accelerometer was initialised to start data recording the following morning at 05:00 AM (day 1), see line 109-110 in the methods. A paragraph addressing the issue of exclusion of more mobile patients has been added to the discussion.

 “Second, we excluded patients with a length of stay ≤48 h and ≤20 h of valid physical ac-tivity measurement. This might have led to the exclusion of patients with higher physical activity levels and less disease severity. However, we measured patients as many days as logistically possible and only included patients in the analysis if they had at least 1 day with ≥20 h out of 24 h of wear time to ensure both day and night measurement.” See line 367-372.

Sedentary behaviour predominant in group

  • Yes, sedentary behaviour is predominant among the patients in our study. However, our observations are in line with other studies exploring the physical activity level in patients hospitalised with community-acquired pneumonia (Rice H, Hill K, Fowler R, et al. Respir Care 2020; Clausen LN, Børgesen M, Ravn P, et al. ERJ Open Res 2019). The predominant sedentarism among patients in our study is mentioned in the discussion.

“Patients admitted with CAP spent over 96% of their in-hospital time in sedentary behaviour, with only 1356 steps/d, which is similar to previous observations.” See line 261-263.

 Note labelling for figures 2 and 3 is confusing and it is difficult to explain why different numbers for the same variables are noted between figure one and two

  • The figure legends for figure 2 and 3 have been moved up just below the figures.

The figure legend for figure 2 has been changed to: “Figure 2. Association between physical activity and sedentary behaviour during admission and risk of readmission and mortality among 166 patients admitted with community-acquired pneumonia. The association between daily step count and physical activity level during admission and risk of 30-day readmission, in-hospital mortality, and 30-day mortality were analysed using unadjusted and adjusted logistic regression models (adjustments: age, sex, and CURB-65 (white)). Steps: Odds ratio per 500-step increase in daily step count during admission. The patients who died in the hospital are not included in the analyses for readmission. LPA: light physical activity, MVPA: moderate-to-vigorous physical activity.” See line 201-208.

The figure legend for figure 3 has been changed to: “Figure 3. Association between physical activity and sedentary behaviour after discharge and risk of readmission and mortality among 77 patients admitted with community-acquired pneumonia. The association between daily step count and physical activity level after discharge and risk of 30-day readmission and mortality were analysed using unadjusted (black) and adjusted logistic regression models (adjustments: age, sex, and CURB-65 (white)). Steps: Odds ratio per 500-step increase in daily step count after discharge. LPA: light physical activity, MVPA: moderate-to-vigorous physical activity.” See line 231-237.

The reason for the different numbers of patients in figure 2 and 3 is that during admission, we have physical activity data from 166 patients, and after discharge, we have physical activity from 77 patients (see figure 1). 

Figure 2. No impact on 30 day readmission  as adjusted rates cancel the effect because of the possible impact of other co morbidities. There is however a significant effect on in hospital and 30 mortality which is only attenuated by adjustment to comorbities

  • Figure 2 shows the association between physical activity during admission and risk of 30-day readmission, in-hospital mortality, and 30-day mortality.

In both the unadjusted (black) and adjusted (white) logistic regression analysis, adjusted for age, sex, and CURB-65 (severity of the pneumonia disease at admission), we did not find any association between physical activity during admission and risk of 30-day readmission.

For both in-hospital and 30-days mortality, we see that increased physical activity during admission is associated with decreased risk of in-hospital and 30-day mortality (unadjusted logistic regression (black)). Even after adjusting for age, sex, and CURB-65, we see the association between increased physical activity during admission and decreased risk of in-hospital and 30-day mortality. 

To further account for disease severity, we did some additional logistic regression analysis, where we, in addition to age, sex, and CURB-65, adjusted for ICU admission, mechanical ventilation, non-invasive ventilation, and high flow therapy. After these adjustments, the models were no longer significant; however, the estimates did not change. See lines 209-226. 

Figure 3. After discharge there is an effect on re-admisson but not on 30 day mortality as these seems to be again to be attenuated by comorbidities

  • Figure 3 shows an unadjusted (black) and adjusted (white) model adjusted for age, sex, and CURB-65 (severity of pneumonia) for the association between physical activity after discharge and risk of 30-day readmission and 30-day mortality. Here we find an association between increased physical activity level after discharge and reduced risk of 30-day readmission, but no association between physical activity level and risk of 30-day mortality. See line 228-246 in the result section.

In the model the steps variable and the sedentary variable may be the ssame variable and seem to mirror each other on the opposite side of the axis,

  • It could seem like the step variable and the sedentary variable are the same variables that mirror each other on the opposite side of the x-axis in the forest plot, but they are not. There is a negative linear association between daily step count and time spent in sedentary behaviour, meaning that the daily step count is reduced for every one percentage point increase in sedentary behaviour.

However, the advantage of showing both variables is that it is possible to calculate the stepping cadence pattern during non-sedentary behaviour. For example, during admission, the patients had a stepping cadence of 27.6 steps per min, whereas after discharge, they had a stepping cadence of 29.7 steps per min.

  • Admission: Non-sedentary behavior: 4.0% = (24 hours * 60 min*4)/100 = 57.6 min;
    1588 steps / 57.6 min = 6 steps/min
  • Discharge: Non-sedentary behavior: 6.2% = (24 hours * 60 min*6.2)/100 = 89.3 min
    2654 steps / 89.3 min = 7 steps/min

Discussion must be qualified. Overall the discussion is confusing, where the main findings form the studies and interspersed with reference from the literature. For clarities' sake, first summarize the positive findings and then progressively show which literature support or disputes such conclusions.

  • The discussion has been rewritten according to the proposition from the reviewer see line 247-378.

The paragraphs are too long and should be more concise explaning one point/finding at a time.

  • As the discussion has been rewritten, we have also paid attention to writing shorter paragraphs, and more paragraphs have been added to the discussion.

While increased exercise may result in a better prognosis, it is also possible that the patients who were less mobile were probably more sick from the pneumonia and the systemic effects of bacteraemia/ septicaemica made them more lethargic more lethargic and as a result more likely to have a worse prognosis. This must be clearly stated in the discussion

  • According to the proposition from the reviewer, a section in the discussion has been added:

    "While physical activity is associated with an improved prognosis, an important issue to address is whether the low physical activity level seen in patients admitted with CAP is simply a marker of disease severity, where severely ill patients are predominantly sedentary, and thus their poor disease outcomes should be attributed to their disease severity alone rather than a combination of disease severity and sedentarism. In a secondary analysis of the association between physical activity during admission and mortality, we included multiple variables (ICU admission, mechanical ventilation, non-invasive ventilation, and high flow therapy) to control for disease severity. After these adjustments, the estimates for the association between physical activity during admission and risk of in-hospital and 30-day mortality remained the same, but models were no longer significant”. See line 277-287 in the discussion.

Over here I see a prospective observational study, while to prove such a hypothesis a randomized trial of standard therapy vs standard therapy plus exercise would be necessary, something which is difficult to organise in acutely sick patients.

  • Our study is a prospective cohort study exploring the association between daily step count and physical activity levels during and after admission with CAP and risk of readmission and mortality. However, to determine whether increased physical activity level during admission improves the prognosis among patients with CAP, randomised controlled trials comparing the effect of standard of care combined with exercise training to standard of care alone are needed. A sentence addressing this issue has been added to the discussion:

 “However, randomised controlled trials comparing the effect of standard of care combined with exercise training to standard of care alone are needed to explore whether an in-creased physical activity level during admission and after discharge improves the prog-nosis among patients with CAP.” See line 330-333.

"In addition, we showed that the risk of in-hospital and 30-day mortality was reduced with an increase in light physical activity and a concurrent reduction in sedentarism during admission in patients with CAP". The authors imply a cause and effect relationship, while this study shows an assocaiation between exercise and better outcome, something which can be explained that more mobile patients are simply less sick. The effect of adjustment on the data tends to favour more the second possibility rather then the first.

  • The sentence has been rewritten to: "Further, we showed that an increase in light physical activity and a concurrent reduction in sedentarism during admission was associated with a 47–53% reduced risk of in-hospital and 30-day mortality." See line 274-277 in the discussion.

"Our results suggest that promoting physical activity through intervention may be required to increase physical activity levels during admission and after discharge to improve the prognosis among patients with CAP." While the word "suggest" and "may" is appropriate it must be clear to the readers, that interventional studies are necessary to prove such a hypothesis.

  • The sentence has been rewritten to ”Our results could suggest that promoting physical activity through intervention may be required to increase physical activity levels during admission and after discharge. However, randomised controlled trials comparing the effect of standard of care combined with exercise training to standard of care alone are needed to explore whether an increased physical activity level during admission and after discharge improves the prognosis among patients with CAP." See line 328-333 in the discussion.

Conclusion: "These findings highlight the need to implement exercise interventions to increase physical activity levels to test the impact on the prognosis of patients admitted with CAP." Disagree with this statement. I think what the authors mean is that interventional studies are needed to support such a hypothesis.

  • The conclusion has been rewritten to: The present study provides evidence that physical inactivity and sedentarism are correlated with prolonged length of stay and increased risk of readmission and mortality in patients admitted with CAP. However, randomised controlled trials are needed to es-tablish whether exercise training initiated during admission improves prognosis among patients with CAP". See line 380-384.

Reviewer 2 Report

Introduction: The age referenced between parenthesis could be eliminated because the sentence provides information to understand the age of the population.

Methods: 

Line 76: “Inclusion criteria were age ≥18 years” The authors must include information in the introduction if young people suffer the same physical inactivity as the elderly. I think it is important to differentiate by age. The functional impairment during the hospital stay was different depending on the age of the patients.  

Line 81-82: Why do you exclude length of stay ≤48 h, or less than one day, with ≤20 h physical activity data during admission?

Line 83-84: The physical activity levels of participants during the lockdown probably were reduced, so do you have taken this aspect into account?

Line 104-105: I understand that the patients should be a minimum of seven days in a hospital stay. If so, I indicate it in the inclusion criteria. 

If the hospital stay is longer than seven days. Did patients wear the accelerometer only seven days during the hospital stays? 

Results

As you indicated in the introduction (lines 44-46) there are some symptoms that reduce physical activity, do you collect? It must be included, such as the characteristics of the participants, in the results.

Author Response

Introduction: The age referenced between parenthesis could be eliminated because the sentence provides information to understand the age of the population.

  • Age references in parenthesis in the introduction and discussion to understand the age of the population in the comparison have been removed.

Methods: Line 76: “Inclusion criteria were age ≥18 years” The authors must include information in the introduction if young people suffer the same physical inactivity as the elderly. I think it is important to differentiate by age. The functional impairment during the hospital stay was different depending on the age of the patients. 

  • The majority of patients in our study were 65 years of age or above (see line 168). Previous studies that have measured physical activity levels among patients hospitalised with community-acquired pneumonia have used the same inclusion criteria with age ≥18 years (Rice H, Hill K, Fowler R, et al. Respir Care 2020; Clausen LN, Børgesen M, Ravn P, et al. ERJ Open Res 2019).

We acknowledge the negative association between age, physical activity level, and functional impairment. Even though pneumonia requiring hospitalisation increases with age and the combined burden of comorbidities, younger patients (aged 18-64) hospitalised with pneumonia are often severely ill and bedridden. Even though previous studies have shown that older individuals are more susceptible than younger to loss of muscle mass (Tanner RE, Brunker LB, Agergaard J, et al. J Physiol 2015). Short-term bed rest in younger healthy individuals still leads to a substantial loss of muscle mass and muscle strength in the lower extremities (Krogh-Madsen R, Thyfault JP, Broholm C, et al. J Appl Physiol (1985) 2010; Nielsen ST, Harder-Lauridsen NM, Benatti FB, et al. Journal of Applied Physiology 2016).

A section addressing this issue has been added in the discussion see line 293-300.

Line 81-82: Why do you exclude length of stay ≤48 h, or less than one day, with ≤20 h physical activity data during admission?

  • In order to be included in the study, the patients should have a length of stay ≥48 hours. Patients were included within the first 24 hours of their admission. After informed consent, the accelerometers were prepared and initialised to begin the recording of physical activity data the following morning at 05:00 AM (day 1). In order to have a valid 24-hour measurement of physical activity during admission, patients need to have worn the accelerometer for at least 20 hours out of a 24-hour measurement period to ensure both day and night measurement. Therefore, patients with a length of stay ≤48 hours would have a too short length of stay in order to measure their physical activity level during admission before they were discharged.

More information on the data collection procedure has been added to the manuscript in the methods see line 107-110: “Patients were instructed to wear the accelerometer for up to 7 consecutive 24 h periods during admission or until discharge (if discharged before day 8) and 7 consecutive 24 h periods after discharge. Data were collected between 05:00 AM on day 1 and 05:00 AM on day 8.”

Further, a paragraph has been added in the discussion addressing the issue of excluding patients with a length of stay ≤48 h see line: “Second, we excluded patients with a length of stay ≤48 h and ≤20 h of valid physical activity measurement. This might have led to the exclusion of patients with higher physical activity levels and less disease severity. However, we measured patients as many days as logistically possible and only included patients in the analysis if they had at least 1 day with ≥20 h out of 24 h of wear time to ensure both day and night measurement..” See line 367-372.

Line 83-84: The physical activity levels of participants during the lockdown probably were reduced, so do you have taken this aspect into account?

  • Patients were included between January 2019 and April 2022. Therefore, some patients were included before the COVID-19 lockdown (from January 2019 until February 2020), and others were included during or after the lockdown (from March 2020 until April 2022).

We acknowledge the fact that the lockdown might have affected the patient's physical activity level. However, in our prospective cohort study (the Surviving Pneumonia Cohort Study), which this study is a part of, we also collected data on the patient's physical activity level prior to admission. The self-reported physical activity level prior to admission was assessed with the international physical activity questionnaire.

We have added data on the patient's physical activity level prior to admission in table 1, line 94-97 in the methods, and line 169-174 in the results. Finally, we made a comparison of the pre-admission physical activity levels between patients included before the lockdown compared to after the lockdown. The comparisons were made with a chi-squared test and did not show any difference in the percentages of patients with a low, moderate, or high physical activity level in the group of patients included before the lockdown compared to during or after the lockdown (Supplementary table 1).

Line 104-105: I understand that the patients should be a minimum of seven days in a hospital stay. If so, I indicate it in the inclusion criteria.

  • In order to be included in the study, the patients should have a length of stay ≥48 h. A length of stay ≥48 h is chosen for practical reasons. Patients are included within 24 h of their admission. Then the accelerometer needed to be prepared and attached to the patient and further have at least 20 h of valid wear time out of 24 h measurement to be included in the study in order to ensure both day and night measurement. The description of accelerometer wear time in line 107-110 in the methods has been changed.

If the hospital stay is longer than seven days. Did patients wear the accelerometer only seven days during the hospital stays?

  • The patients wore the accelerometer for up to 7 days during admission (from 5:00 AM on day 1 until 05:00 AM on day 8). If the patient was discharged before day 8, they wore the accelerometer until discharge. If the patient was hospitalised for longer than 8 days, the accelerometer was removed after the accelerometer stopped collecting data on day 8. See line 107-110 in the methods.

Results: As you indicated in the introduction (lines 44-46) there are some symptoms that reduce physical activity, do you collect? It must be included, such as the characteristics of the participants, in the results.

  • Data on oxygen supplementation and intravenous antibiotic treatment have been added to the baseline characteristics Table 1 and the results section line 175-176.

Reviewer 3 Report

In this manuscript, the authors conducted a prospective cohort  study to investigate the impact of physical inactivity and sedentarism during and after admission on the prognosis of patients admitted with community-acquired pneumonia (CAP), including length of hospital stay, 30-day readmission, in-hospital and 30-day mortality. The study provides valuable information to clinical practice regarding CAP patient management. The study design and methods are appropriate, and the manuscript is well written. I have the following comments:

1   1) In Table 1, the authors described the baseline characteristics of the patient cohort including the number of comorbidities. Could the authors provide some basic information about those comorbidities? For instance, can the authors list the major comorbidities the patients had in the table or in a footnote under the table? Such information would help readers better understand the patient cohort.

1    2)  Line 160-161: The authors presented data on mean wear time of accelerometers during admission (4.0+2.2 days) and after discharge (6.3+1.6 days). Is it possible to provide the percentage of the total days that all the patients stayed in hospital being wear time? Similarly, what is the percentage of the total follow-up days of the 77 patients with activity data after discharge being wear time?   

3   3) Line 170-171: It is stated that “Length of stay was reduced by 6.6% (95% CI 2.0-10.9, p<0.01) for every 500-step increase in daily step count on day one”. Can the authors clarify whether these results are from a multivariable analysis, and if so, provide the variables that are adjusted for? Lastly, is there a typo of 95% CI? Should it be 2.0-10.9% instead?

4   4) In Table 1, out of the 166 patients 13 had in-hospital mortality and 16 had 30-day mortality. Do the 16 30-day deaths include the 13 in-hospital deaths? Please clarify. If they do, then it means that among the 77 patients with activity data after discharge, there were at most three 30-day deaths. If this is true, then I am afraid the statistical power was too low for the analysis of the association between physical activity after discharge and 30-day mortality.

5   5) In the study, the authors did not find any association between physical activity (including daily step count) during admission and risk of 30-day readmission, but they found most of these physical activity factors after discharge were associated with 30-day readmission. Could the authors provide a possible reason for it?

6   6) A couple of data discrepancies are found in Table 1. In Table 1, there were totally 156 (=78+58+20) patients with CURB-65 data, and 11 patients with CURB-65 data missing, leading to a total of 167 patients, not 166. In addition, out of the 166 patients 37 had 30-day readmission, which is 22.3% of all the patients , not 24.2%.

Author Response

In this manuscript, the authors conducted a prospective cohort study to investigate the impact of physical inactivity and sedentarism during and after admission on the prognosis of patients admitted with community-acquired pneumonia (CAP), including length of hospital stay, 30-day readmission, in-hospital and 30-day mortality. The study provides valuable information to clinical practice regarding CAP patient management. The study design and methods are appropriate, and the manuscript is well written. I have the following comments:

In Table 1, the authors described the baseline characteristics of the patient cohort including the number of comorbidities. Could the authors provide some basic information about those comorbidities? For instance, can the authors list the major comorbidities the patients had in the table or in a footnote under the table? Such information would help readers better understand the patient cohort.

  • Information about the 9 major comorbidities has been added to Table 1.

 Line 160-161: The authors presented data on mean wear time of accelerometers during admission (4.0+2.2 days) and after discharge (6.3+1.6 days). Is it possible to provide the percentage of the total days that all the patients stayed in hospital being wear time? Similarly, what is the percentage of the total follow-up days of the 77 patients with activity data after discharge being wear time?  

  • Information on the mean percentage wear time during admission and after discharge has been added in the results section. See line 184-186.

Line 170-171: It is stated that “Length of stay was reduced by 6.6% (95% CI 2.0-10.9, p<0.01) for every 500-step increase in daily step count on day one”. Can the authors clarify whether these results are from a multivariable analysis, and if so, provide the variables that are adjusted for? Lastly, is there a typo of 95% CI? Should it be 2.0-10.9% instead?

  • The typo mistake has been corrected and information on the type of analysis has been added to the manuscript. See line 195.

“In a multivariable analysis adjusted for age, sex, and CURB-65, length of stay was reduced by 6.6% (95% CI 2.0–10.9%; p<0.01) for every 500-step increase in daily step count on day 1".

In Table 1, out of the 166 patients 13 had in-hospital mortality and 16 had 30-day mortality. Do the 16 30-day deaths include the 13 in-hospital deaths? Please clarify. If they do, then it means that among the 77 patients with activity data after discharge, there were at most three 30-day deaths. If this is true, then I am afraid the statistical power was too low for the analysis of the association between physical activity after discharge and 30-day mortality.

  • In figure 3, it is correct that there are only 2 patients out of the 77 patients with activity data after discharge who dies during the 30-day follow-up period from discharge. Therefore, unsurprisingly, we did not find any statistically significant association. However, we still think it is useful to report the odds ratios and the corresponding confidence intervals (which are wide) to provide a quantification of the magnitude of the association. A paragraph addressing this issue has been added to the discussion, see line 310-313 in the discussion.

In the study, the authors did not find any association between physical activity (including daily step count) during admission and risk of 30-day readmission, but they found most of these physical activity factors after discharge were associated with 30-day readmission. Could the authors provide a possible reason for it?

  • In the study, we did not find any association between physical activity during admission and risk of 30-day readmission. However, we found that increased physical activity level after discharge was associated with a reduced risk of 30-day readmission.

From our data, it seems like physical activity level during admission is more related to whether or not the patient survives their admission and is discharged alive. Whereas physical activity level after discharge is related to the patient's recovery after discharge and whether readmission can be prevented. A paragraph addressing this issue has been added to the discussion see line 313-316.

A couple of data discrepancies are found in Table 1. In Table 1, there were totally 156 (=78+58+20) patients with CURB-65 data, and 11 patients with CURB-65 data missing, leading to a total of 167 patients, not 166. In addition, out of the 166 patients 37 had 30-day readmission, which is 22.3% of all the patients, not 24.2%.

  • We thank the reviewer for highlighting the discrepancies found in Table 1. The mistake in the data on CURB-65 has been corrected. In regard to data on 30-day readmission, the stated percentage is correct as it is the number of patients who are discharged alive, who are readmitted. So the number is 37 patients out of 153 (166 patients minus 13 patients who died during admission), giving 24.2%.

Round 2

Reviewer 1 Report

paper is much improved